

# Communication behavior recognition using CNN-based signal analysis

Hao Meng[1], Yingke Lei[1], Fei Teng[1], Jin Wang[1], Changming Liu[2] and Caiyi Lou[3]

[1] School of Electronic Countermeasures, National University of Defense Technology, Anhui, China
[2] Electronic Countermeasures Division, Tongfang Electronic Technology Co., Ltd., Jiangxi, China
[3] Jiaxing Communication Information Security Control Laboratory, Technology Research Institute, Anhui, China

## ABSTRACT

This article explores the technology of recognizing non-cooperative communication behavior, with a specific emphasis on analyzing communication station signals. Conventional techniques for analyzing signal data frames to determine their identity, while precise, do not have the ability to operate in real-time. In order to tackle this issue, we developed a pragmatic architecture for recognizing communication behavior and a system based on polling. The method utilizes a one-dimensional convolutional neural network (CNN) to segment data, hence improving its ability to recognize various communication activities. The study assesses the reliability of CNN in several real-world scenarios, examining its accuracy in the presence of noise interference, varying lengths of interception signals, interferences at different frequency points, and dynamic changes in outpost locations. The experimental results confirm the efficacy and dependability of the convolutional neural network in recognizing communication behavior in various contexts.

## INTRODUCTION

Within the domain of real-time and predictive disciplines, traditional approaches frequently encounter limitations that restrict their effectiveness. As a workaround to these restrictions, the recognition of communication behavior at the signal level has emerged as a potential approach. A particularly notable aspect of this topic is the examination of how radiation sources communicate, which is a primary focus in the wider field of radio cognition. This involves extracting features from intercepted signals released by communication radiation sources and analyzing the unique radio frequency characteristics present in the signals to understand the underlying communication behavior (*Khalid & Anpalagan, 2010*). The conventional approach to making this determination involves analyzing discrepancies in the radio frequency spectrum, either among signals transmitted by different individuals engaged in the same communication behavior or among signals generated during distinct communication activities. The incorporation of this research with existing information forms the basis for identifying the communication behavior of radiation sources.

Corresponding author
Caiyi Lou, 13203155711@163.com

In the radio frequency domain, different signal-emitting entities operate and communicate in a variety of ways, which causes signal sequences to be sent with different properties (*Liu et al., 2016*; *Hu et al., 2019*; *Xu et al., 2023a*). This variance allows for the analysis of communication behavior associated with a signal, without the need to examine the internal structure of the signal itself. This article investigates the transmission characteristics of a specific type of emitter communication method. It examines communication data acquired through time-domain or spectral space monitoring in order to gain understanding and comprehend these unique characteristics (*Axell, Leus & Larsson, 2010*; *Cao et al., 2021*).

Convolutional neural networks (CNNs) have demonstrated exceptional effectiveness in the field of image recognition and processing during the present era of data-driven research. The question arises as to whether these neural networks can effectively utilize their achievements to classify and identify one-dimensional time-domain data, especially when obtained through mid-frequency sampling (*Chen et al., 2023*). The objective of this study is to investigate this topic by developing a system that can detect communication behaviors using data obtained from a communication station. The main goal is to clarify a methodology for defining behavior, create a customized convolutional neural network for this purpose, and empirically confirm its effectiveness. Another goal is to provide in-depth understanding of the application of advanced neural network structures for detecting communication patterns in radiation source signals.

## BACKGROUND

The radiation source discussed in this study is essential for establishing communication links between air and ground interfaces, as well as ships, aircraft, fleets, and land-based stations. This facilitates the transfer of information smoothly and effortlessly between these interfaces. This radiation source facilitates the exchange of digital information between terminals by leveraging network communication technology and adhering to a standardized message format. The radiation station, referred to as Model A, serves as a crucial element in enabling standardized and efficient communication protocols across various interfaces and domains. The communication scenario employed in this article involves a polling communication scenario with multiple targets, one master station, and multiple slave stations.

### Operating modes of the emitter network

The network of the Model A communication station is specifically intended to enable digitalized ship-to-air command and thorough sharing of situational information. Model A emitter primarily utilizes a polling operating strategy to do this (*Chen et al., 2022*; *Cao et al., 2022*; *Xu et al., 2023b*). In this arrangement, the network control station functions as the primary station, systematically querying the subordinate stations (outposts). Consequently, data from these subordinate stations is transferred sequentially, promoting accurate digital interactive communication within the context of the "one center, multiple slave stations" model.

| call message structure of network control station | Head Message | Phase Reference | Outpost Address Code | | | |
|---|---|---|---|---|---|---|

| Outpost Response Message Format | Head Message | Phase Reference | Start Code | Variable Data | End Code | |
|---|---|---|---|---|---|---|

| Network Control Station Report Message Format | Head Message | Phase Reference | Start Code | Variable Data | End Code | Outpost Address Code |
|---|---|---|---|---|---|---|

**Figure 1** Polling call message format frame structure.

The architecture of the Model A communication station consists of five main operational modes: polling call, network synchronization, short broadcast, long broadcast, and radio quiet. The subsequence sections outline each operational mode separately and clarify their unique functionalities and operational techniques (*Qiao, Wang & Gao, 2020*). Within the communication network, the overall system operates using a polling mechanism, where there is one master station and multiple slave stations. The system utilizes network synchronization and short broadcast signal working modes to synchronize information and time across the entire network. Additionally, it employs long broadcast signals for overall macro control.

### Polling call

The polling call is a conventional operating method used in the communication network of the Model A communication station. In this arrangement, a network control station plays a central role, while additional network access devices are designated as outposts. The network control station functions as the central hub, coordinating the whole network by establishing a sequential polling call for all outposts. Every outpost communicates relevant data and information during its designated time window (*Zheng et al., 2017*).

The polling call mechanism is crucial for ensuring that tactical information and data are effectively shared among all network-connected units within a particular range, maximizing the efficiency of network usage.

Three different message formats are used during polling calls (*Ma et al., 2023*): the call information format of the network control station, the response information format of the outpost, and the report information format of the network control station, as shown in Fig. 1.

When a message is sent, it goes through a process that includes the transmission of five header frames, followed by one phase reference frame. The purpose of the five header frames is to establish frame synchronization, while the phase reference frame assists in the measurement and calibration of the Doppler frequency shift. The calibration process establishes the zero-phase reference point and efficiently decreases the bit error rate of the signal during analysis. In addition, when transmitting information that contains data, it is necessary to include two frame start codes before the data and an end code after the data to identify the end of the transmission.

### Network synchronization

The main objective of the network synchronization mode is to establish a uniform time reference throughout the communication network of the Model A communication station. During network synchronization, only the network control station is operational over the whole communication network. The process is initiated by the network control station through the transmission of a synchronization message, which includes a header frame consisting of five frames. This message mostly consists of Doppler tones and synchronization signals, notably incorporating two unique tone signals: 605 Hz and 2,915 Hz (*Fang et al., 2013*).

### Short broadcast

During manual operation mode, an operator uses a network access unit, such as a network control station or an outpost, to send a single data report, known as a brief broadcast message, to other network access units. After being transmitted, the unit smoothly switches to the receiving state. The format used for the brief broadcast message closely resembles that of the reply message usually sent from the outpost (Fig. 1).

### Long broadcast

When important conditions arise or when urgent information needs to be sent, an access unit in the communication network enables its operator to consistently send a single data report to other network members. This continuous transmission consists of a lengthy broadcast message, consisting of a series of brief broadcasts interspersed with two frames between each short broadcast message. The operator starts the extended broadcast message, which remains active until manually halted. When the system stops, the main operator takes action and changes the system to a different mode of operation.

### Radio silence

During a period of radio silence, the network access unit functions exclusively in a passive mode, where it only receives data from other network access units and does not broadcast any information. It remains unresponsive despite being prompted by the network control station.

## Signal format
### Signal frame

The signal transmitted by the Model A communication station is a sophisticated, multi-tone synthetic signal. The system consists of 16 separate audio points that are used for signal synthesis. These points are described in detail in Table 1. Additionally, all transmissions take place inside frames. Every frame is specifically engineered to accommodate 24 bits of data for the purpose of transmitting tactical information. The (30,24) Hamming code is used to convert a 24-bit data into a 30-bit signal. This signal includes a 6-bit check code that is crucial for detecting and verifying errors.

The Model A communication station signal consists of multiple components, including the header frame, phase reference frame, data frame, and other types. All of these components are modulated using the DQPSK modulation scheme (*Alhamad & Boujemaa, 2019*).

**Table 1  Signal frequency points of model A communication station.**

| Number | Frequency (Hz) | Purpose | Bit | Number | Frequency (Hz) | Purpose | Bit |
|---|---|---|---|---|---|---|---|
| 1 | 605 | Doppler shift | | 9 | 1,705 | | 14, 15 |
| 2 | 935 | | 0, 1 | 10 | 1,815 | | 16, 17 |
| 3 | 1,045 | | 2, 3 | 11 | 1,925 | | 18, 19 |
| 4 | 1,155 | | 4, 5 | 12 | 2,035 | | 20, 21 |
| 5 | 1,265 | Data | 6, 7 | 13 | 2,145 | Data | 22, 23 |
| 6 | 1,375 | | 8, 9 | 14 | 2,255 | | 24, 25 |
| 7 | 1,485 | | 10, 11 | 15 | 2,365 | | 26, 27 |
| 8 | 1,595 | | 12, 13 | 16 | 2,915 | | 28, 29 |

### Signal structure

The signal emitted by the Model A communication station is organized into frames, which consist of separate elements: the header frame, phase reference frame, control code, and message data. The format and duration of the header frame, phase reference frame, and control code stay consistent, while the length of the message contents fluctuates depending on the transmitted message.

**Header frame**: The header frame serves as a binary signal composed of two distinct frequencies, namely 605 Hz and 2,915 Hz. The main objective of the device is to coordinate the reception and correct any Doppler frequency shifts for the network access unit. The transmission of every message commences with the emission of five consecutive header frames.

**Phase reference frame**: The phase reference frame directly follows the header frame. This frame utilizes the QPSK modulation technique and includes all 16 frequency points. Its purpose is to provide the zero point for the phase reference in the next frames. This facilitates the reduction of the bit error rate in data tones.

**Control code**: The control code consists of three essential codes: the start code, stop code, and address code. The code consists of two frames that resemble the previously described data frames. These frames communicate specialized information, such as start code, stop code, and address code, using 16 single-tone modulated synthetic multi-tone signals. Therefore, the control code's constituent frame also serves as a data frame.

- *Start code* is observed 2 frames after the phase reference frame, signifying the beginning of the message body.
- *Stop code* appears 2 frames after the end of the message body, indicating the end of the message body.
- *Address code* is a system comprised of two frames that is used to indicate the current outpost address in response to a call from the network management site. The varied addresses of outpost stations enable the recognition of behavior using these codes, providing the potential to identify communication activities.

**Message information**: The message data encapsulates the communicated information, extending from the start to the stop codes. The number of data frames varies depending on the content of the data.

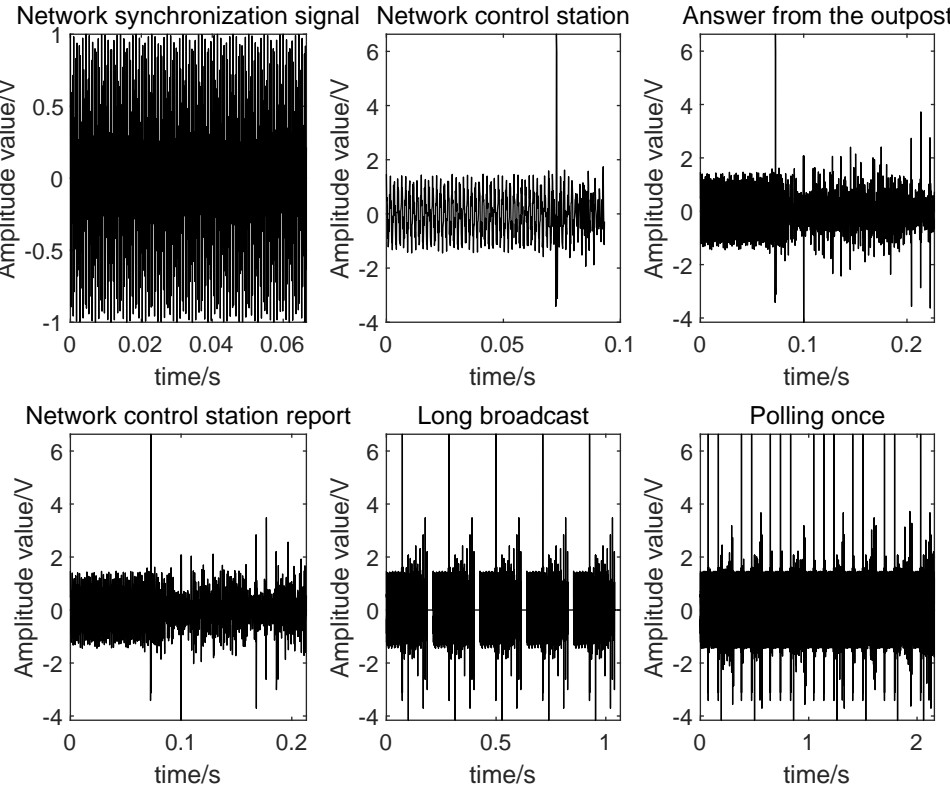

**Figure 2** Link-11 tactical data link signal simulation.

Based on the aforementioned descriptions, the simulated signals for each operational mode are depicted in Fig. 2.

## Network simulation

Our first step is to develop a communication network so that we can investigate the communication behavior recognition of the Model A communication station-based network. In order to produce real communication signals and data for later learning and recognition operations, this network development is necessary.

In light of the Model A communication station's signal protocol, which was previously described and mostly relies on polling, we developed a "single-center mesh structure." This organization is centered on a network control station that branches out into several unit outposts. Each outpost has a distinct location code that changes dynamically in accordance with its actual physical location (*Luo et al., 2019*). Pre-established communication protocols are used to coordinate communication between stations.

Additionally, this article adds components like single-point interference and channel Gaussian white noise to as closely mimic real-world conditions as feasible. These augmentations imitate the possible use of interference on specific frequency points by

non-cooperative parties. Furthermore, station movement amplitude and speed variations are simulated to reflect changes in the surrounding environment.

With the purpose of developing a strong framework for the investigation of communication behavior recognition in this network environment, a scenario for identifying communication networks, including the Model A communication station, is developed using these criteria.

## Modeling of communication behavior

Our work focuses on the implementation of recognition algorithms to identify various communication behaviors in model A communication stations, specifically in the context of communication behavior recognition technology. We utilize convolutional neural networks for classification and prediction. It identifies specific communication patterns by analyzing differences in time, frequency, and spatial characteristics in intercepted receiver data. The first phase is to concentrate on creating a communication scenario, followed by describing the communication behavior that is fundamental to model A communication stations. The following provides further details on the definition of these diverse communication behaviors.

**Polling**: Polling is the primary operating method used in the network of a model A communication station. Therefore, the determination of communication behaviors for each network is dependent on the number of network control stations and outposts. For example, let's consider a case where there is one network control station, referred to as N, and two outposts, A and B (the number of stations is not relevant for this example). The communication behavior of the network polling is as follows:

1. N calls A, A answers N
2. N calls B, B answers N
3. N calls A, A does not answer N
4. N calls B, B does not answer N
5. N conducts network control station report

We are examining the possibility of recognizing the polling communication behavior as previously defined, namely at the signal level. The following is a qualitative analysis of the feasibility of this delineation strategy.

Considering the identification of the five specified polling communication behaviors, behaviors 1 and 2 demonstrate noticeable signal frame patterns linked to network control station calls and outpost answers. As a result, these behaviors may be easily differentiated from the five stated earlier. Behaviors 3 and 4 demonstrate distinct frame structures specifically designed for network control station calls, whereas behavior 5 exclusively offers frame structures for network control station reports. Furthermore, behaviors 1 and 3 correspond to the address code of outpost A, whereas behaviors 2 and 4 correspond to the address code of outpost B. Therefore, it seems possible to differentiate the aforementioned five behaviors based on a signal perspective.

**Network synchronization**: Network synchronization refers to the process in which the network control station sends synchronization signals at regular intervals to correct the time reference of the whole network. This process is the only way to achieve network

synchronization. The network synchronization signal is clearly identified due to its peculiar frame structure (*Chen et al., 2021*).

**Long broadcast**: The extended broadcast signal comprises many brief broadcast signals, with a gap of two frames between each one. During a prolonged broadcasting situation, each brief broadcast signal remains unchanged and functions autonomously from the others. The frame structure of the response signal of the outpost is mirrored by its frame structure.

# METHOD

## Signal preprocessing

Considering the output of the model's signal A communication station employs FM modulation in the HF or UHF band for transmission. The operational technique involves first demodulating the intercepted signal to extract the modulation information. Afterwards, the obtained signal is subjected to sampling. In order to adhere to Nyquist sampling requirements and optimize algorithmic efficiency, a sampling frequency of 7,000 bps is chosen, taking into account the maximum signal frequency of 2,915 Hz. This choice guarantees the convenience of computation without sacrificing the final result.

Our objective is to generate signal datasets that represent a variety of communication behaviors. In addition to determining the corresponding tag values and establishing a direct correlation between the sampling set of the communication signal and the particular communication behavior, our objective is to accomplish this. The objective is to enhance subsequent learning and recognition of CNN networks. How can a collection of communication behavior signals be constructed? Consider, for instance, the previously mentioned polling communication behavior 1: "N calls A, A answers N", the procedure proceeds as follows:

- Step 1: Identify the involved sites in the communication behavior, namely network control station N and outpost A.
- Step 2: Configure the communication network as outlined previously and simulate the continuous behavior 1 accordingly.
- Step 3: The receiver continually retrieves intercepted data of behavior 1, conducts demodulation and potential sampling, resulting in a collection of signal datasets that encompass diverse contents yet pertain to behavior 1, amassed over an extensive duration.
- Step 4: Segment the obtained dataset based on the prescribed length $N$ (assuming the length as $N$, where $N$ is any positive integer). This value can be adjusted concerning the signal interception period inputted into the CNN network (*Guo et al., 2022*). The schematic diagram of segmentation is shown in Fig. 3. As the rated segmentation length might not consistently match the number of sampling points of a singular communication behavior, this method enables interception at any behavior's starting time while averting randomness.
- Step 5: Continue segmenting the simulated signal until the remaining sampling points are fewer than the rated length $N$, resulting in $M$ segmentation groups. This process yields the summary matrix $H_{M \times N}$, corresponding to behavior 1.

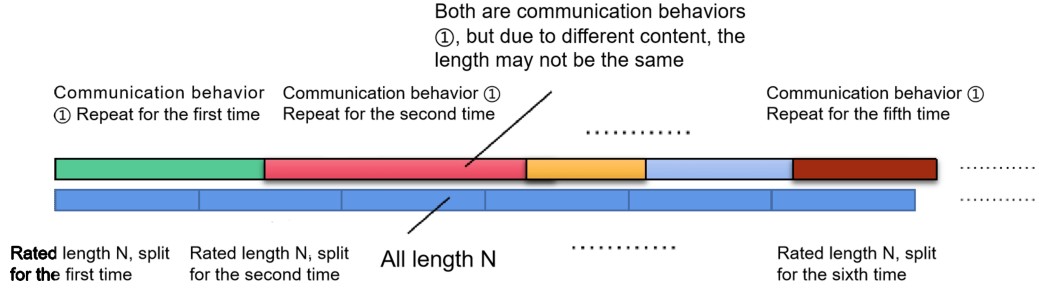

**Figure 3** Schematic diagram of intercepted data set segmentation method.

The aforementioned steps outline the process of attaining the summary matrix $H$, representing the signal dataset aligned with the communication behavior. These operations are conducted across all network communication behaviors, culminating in signal preprocessing—the construction of the CNN input dataset—where each row vector of the summary matrix $H$ is formed.

## Communication behavior recognition algorithm
### Design of the CNN

The parameters of each component in this specific topic are customized based on the fundamental properties and functionalities of convolutional layers, pooling layers, fully connected layers, and activation functions inside convolutional neural networks.

The algorithm utilizes a one-dimensional array obtained from sampling and processing the genuine communication behavior signal as the input matrix. Like the preprocessing stage, the convolution kernel is a one-dimensional row vector. In order to enhance the extraction of subtle characteristics while simplifying the algorithm's complexity, and taking into account the convenience of aligning with odd-sized kernels, the decision is made to utilize the smallest odd-sized convolution kernel, which is $1 \times 3$ (*Soydaner, 2022*).

Choosing the average pooling approach prioritizes the contextual information in the data, which may restrict the extraction of texture characteristics, particularly edge information, from the sample data. This article utilizes the maximum pooling method for pooling, as one-dimensional input data matrices inherently contain important edge properties. Our preliminary experiments demonstrated that using the smallest $1 \times 2$ pooling kernel can effectively retain channel information while reducing algorithm complexity and the number of network parameters. Prior to pooling, a set of parameters with a size of $\frac{N}{2} \times \frac{N}{2}$ is used in order to further control complexity.

After the maximum pooling layer, a batch standardization layer is added to improve the speed of the algorithm and accelerate the training process before entering the fully linked layer (*Sun et al., 2020*). After reducing the dimensions, the network next applies the softmax function to classify and generate the communication behavior recognition matrix, which represents the final classification result. The configuration of two sets consisting of convolution and pooling layers corresponds to the computational parameters described in this research.

*Process of communication behavior recognition*

This article examines a simulated communication scenario called the "single network control station, three outpost stations" configuration. It identifies seven specific communication behaviors, including polling, network synchronization, and extended broadcasts. Every behavior is associated with sets of signal samples, resulting in a total of sets of communication signal samples for Model A communication stations. The signal sets are essential data used to train and validate the algorithm for recognizing communication behavior in Model A communication stations, which is based on convolutional neural networks. The essential stages of the cognitive algorithm are as below.

- Step 1: Employ the established communication network to simulate communication behavior and intercept the resultant communication dataset.
- Step 2: Demodulate the intercepted data, perform sampling at intermediate frequencies, preprocess the data, and segment the signal to generate a matrix denoted as $7 \times M$ comprising Link-11 data link signal samples $S_{7M \times N} = [s_1; s_2; ...; s_k; ...; s_{7M \times N}]$;
- Step 3: Randomly partition the signal sample matrix $S_{7M \times N}$ from Type A communication stations into a training set, $S_{tr}$ and a test set, $S_{te}$;
- Step 4: Randomly designate 20% of the training set as the validation set $S_{va}$, while the remaining 80% compose the updated training set $S_{tr}$. This updated set $S_{tr}$ is utilized for CNN model training, leveraging $S_{va}$ to fine-tune the model's local parameters.
- Step 5: Utilize the trained network model to classify communication behaviors using the test set $S_{te}$;
- Output: Classification outcomes of the seven communication behaviors exhibited by the communication station.

## EXPERIMENTAL RESULTS AND DISCUSSION

To evaluate the efficacy of the CNN algorithm in recognizing communication behavior and its adaptability across various scenarios, a diverse set of signal datasets were gathered. The datasets were utilized to assess the algorithm's feasibility in various external circumstances. Specifically, changes were made to four external factors: the noise level, the duration of intercepted signals, the frequency points used for interference suppression, and the dynamic shifts in sentinel positions. The algorithm's performance was thoroughly evaluated and scrutinized through these enhancements.

### Impact of noise intensity

The open channel facilitates the reception and detection of signals from the non-cooperative Model A communication station. Nevertheless, this transparency also allows for possible disruptions to our operations, which can adversely affect the accuracy of the algorithm's ability to recognize communication activity, as discussed in this article. The purpose of this part is to assess and confirm the ability of the convolutional network recognition algorithms to withstand noise.

Here, we propose a method using a controlled variable approach, in which all conditions are kept constant except for the varying degree of noise interference. This modification

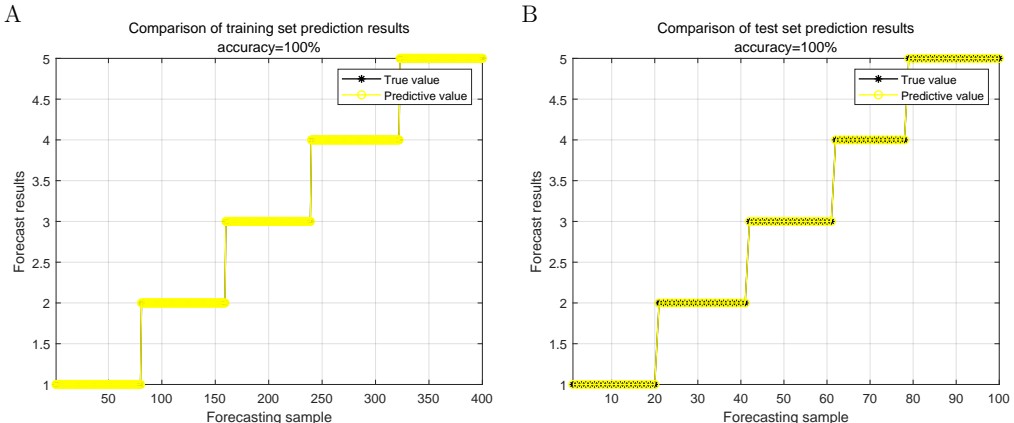

**Figure 4  Recognition rate without noise interference.**

enables us to examine the impact of varying levels of noise interference on the rate at which communication behavior is recognized.

To account for the random nature of Gaussian signal white noise, the Monte Carlo approach is used to repeatedly perform recognition rate studies on several test sample sets, each with different levels of noise interference. The mean recognition rate is subsequently calculated. We conduct correlation analysis to generate precise results by plotting a graph that illustrates the relationship between noise interference intensity and communication behavior recognition rate.

At first, the control group is subjected to a test to determine the rate at which they recognize communication behaviors. This test uses a sample set that is free from any interference or disturbances. This control set functions as a standard for evaluating and contrasting with other test sample sets that contain noise. Figure 4 depicts the process of learning and recognizing communication behavior using the algorithm. It shows the accuracy of the algorithm in recognizing communication behavior on a test sample set without any noise. Figure 4 clearly illustrates that the identification rate steadily increases, exhibiting a conspicuous trend and eventually reaching 100%.

Various experimental groups were established, each with varied levels of noise interference intensity. The signal-to-noise ratios for these groups were set at 0 dB, −0.69 dB, −1.09 dB, −1.38 dB, −1.60 dB, −1.79 dB, −1.94 dB, and −2.07 dB, respectively. The number of tests conducted in each group was multiplied by 60, and the identification rates of communication behaviors with varying signal-to-noise ratios were then computed. These noise conditions are set to be 1 and 1.5 times the signal power, respectively.

## Impact of signal interception time

Obtaining a complete signal from a non-cooperative source is difficult due to the limitations of non-cooperative interception. Interference and anti-interception mechanisms frequently restrict the length of uninterrupted interception, leading to an occasionally disrupted

**Table 2  Recognition rate corresponding to different interception durations.**

| Test set capture duration (seconds) | Recognition accuracy | Test set capture duration (seconds) | Recognition accuracy |
|---|---|---|---|
| 1 | 100 | 0.084 | 23.6 |
| 0.5 | 100 | 0.083 | 25.9 |
| 0.133 | 100 | 0.082 | 21.6 |
| 0.12 | 100 | 0.081 | 25.3 |
| 0.1 | 100 | 0.08 | 24.1 |
| 0.095 | 98.7 | 0.075 | 27.2 |
| 0.09 | 99.3 | 0.07 | 23.9 |
| 0.088 | 97.1 | 0.06 | 22.6 |
| 0.087 | 91.4 | 0.05 | 21.7 |
| 0.086 | 89.1 | 0.03 | 25.2 |
| 0.085 | 74.7 | 0.01 | 26.1 |

intercepted signal. The intricacies involved present substantial obstacles to the real-time efficiency of algorithms designed to recognize communication activity.

Therefore, it is essential to ascertain the time of uninterrupted signal interception necessary for our algorithm to satisfy recognition rate criteria and practical application standards. In this section, controlled variables are used to keep other aspects constant while constructing distinct test sample sets with varying lengths of intercepted signals. To limit the effects of chance, recognition rate testing trials are regularly done across diverse test sample sets using the Monte Carlo approach. The subsequent mean recognition rate is computed.

A total of twenty-two unique test sample sets, each with varying interception durations, were created for study. The recognition rates were then computed for each individual sample set. The average recognition rates determined from these sets are presented in Table 2. Additional analysis involves creating a graph that illustrates the connection between the duration of intercepted signals and the rate at which communication behavior is recognized. This is then followed by correlation analyses to draw precise results.

Using the data obtained from Table 2, Fig. 5 was created to show the relationship between the length of intercepted signals and the accuracy of recognizing communication patterns.

Figure 5 demonstrates that there is no significant correlation between the duration of intercepted signals and the recognition rate of communication behaviors. Significantly, when the duration of the intercepted signal surpasses 0.085 s, the accuracy of recognition remains stable at approximately 99%, indicating consistent and reliable performance. During this time period, the algorithm demonstrates outstanding recognition abilities and consistently achieves a high level of accuracy. Nevertheless, if the duration of the intercepted signal is less than 0.085 s, indicating shorter periods, the rate at which it can be recognized decreases significantly to around 23%. For each type of signal duration, we repeated the test 60 times, setting the number of samples at 50 for each type. The algorithm's capacity to recognize is greatly reduced at such short durations, making the duration threshold

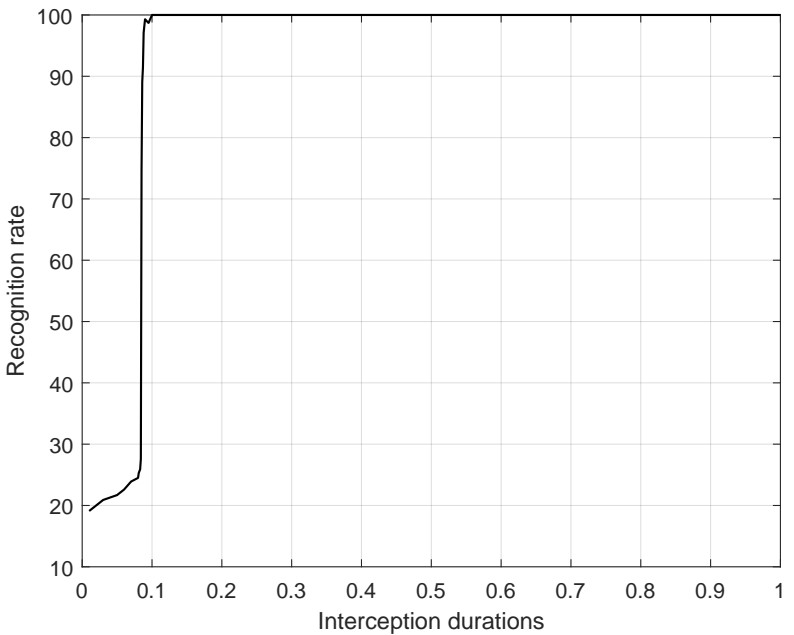

**Figure 5** Recognition rate corresponding to different interception durations.

of 0.085 s the minimum reliable limit for recognition by this communication behavior recognition method.

## Impact of outpost movement

The method faces dynamic networks in real-world scenarios, which are characterized by frequent changes in physical or network addresses. These changes lead to fluctuations in address codes at the signal level. In order to ensure compatibility with real-world applications, it is crucial to assess the flexibility of the one-dimensional CNN-based algorithm in dynamic network scenarios (*Chu, Xiao & Liang, 2020*).

The address code's bit count underwent a progressive transition, ranging from 1 to 48, covering the complete range of variations in address code size from tiny to large. Due to the intrinsic randomness of these modifications, each modification does not consistently correlate to the same code, even when the total number of bits remains constant. In order to reduce the impact of this unpredictability, the Monte Carlo approach is utilized to examine this situation. For each type of signal duration, we repeated the test 60 times, setting the number of samples at 50 for each type. Figure 6 illustrates the resulting curve.

The calculated linear correlation coefficient between the two variables is $-0.9099$, showing a strong negative association. Therefore, it is clear that changes in the extent of outpost movement have a direct impact on the rate at which communication behavior is recognized, with a constant decrease observed as the extent of movement increases. This is consistent with our comprehension and empirical observations.

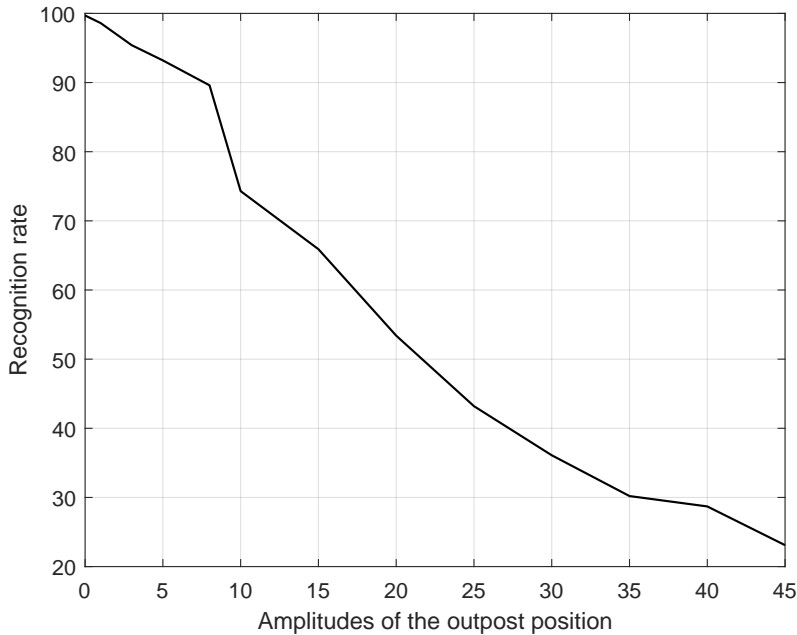

**Figure 6** **Recognition rate corresponding to different amplitudes of the outpost position.**

## CONCLUSIONS

The real-time accuracy of the emergent cognitive technology in non-cooperative tactical data link communication behavior, which focuses on physical layer information, is attracting attention. By examining communication behavior recognition through the lens of a communication station's signal set, this article verifies the viability of a recognition algorithm based on a one-dimensional convolutional neural network and provides a methodical approach to this technology. The research findings indicate that the algorithm successfully classifies communication behavior and adjusts to diverse environments, thereby satisfying the criteria of real-time performance and universality. However, beyond that, the study solely presents a convolutional neural network for recognition and classification and establishes a foundational cognitive framework. Further research and verification are required to effectively implement this technology in practical applications.

### Funding

The study was supported by the National University of Defence and Technology (Project number: 62071479). There was no additional external funding received for this study. The funders had no role in study design, data collection and analysis, decision to publish, or preparation of the manuscript.

## Grant Disclosures

The following grant information was disclosed by the authors:
The National University of Defence and Technology: 62071479.

## Competing Interests

Changming Liu is employed by Electronic Countermeasures Division, Tongfang Electronic Technology Co., Ltd., Jiangxi, China. Other authors declare that they have no competing interests.

## Author Contributions

- Hao Meng conceived and designed the experiments, performed the experiments, analyzed the data, performed the computation work, prepared figures and/or tables, authored or reviewed drafts of the article, and approved the final draft.
- Yingke Lei performed the experiments, analyzed the data, authored or reviewed drafts of the article, and approved the final draft.
- Fei Teng performed the experiments, analyzed the data, authored or reviewed drafts of the article, and approved the final draft.
- Jin Wang performed the experiments, analyzed the data, authored or reviewed drafts of the article, and approved the final draft.
- Changming Liu conceived and designed the experiments, authored or reviewed drafts of the article, and approved the final draft.
- Caiyi Lou conceived and designed the experiments, authored or reviewed drafts of the article, and approved the final draft.

## Data Availability

The MATLAB code used in the study and the raw data are available in the Supplemental Files.

## Supplemental Information

Supplemental information for this article can be found online at http://dx.doi.org/10.7717/peerj-cs.2036#supplemental-information.

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
