# Peer review of "Communication behavior recognition using CNN-based signal analysis"

_PeerJ Computer Science, doi:10.7717/peerj-cs.2036_

## Round 0.1 · original submission · Major Revisions

Please consider the feedback provided by the reviewers and revise the manuscript accordingly, focusing particularly on enhancing the clarity of the language and the experiments.

**Language Note:** The review process has identified that the English language must be improved. PeerJ can provide language editing services - please contact us at [email protected] for pricing (be sure to provide your manuscript number and title). Alternatively, you should make your own arrangements to improve the language quality and provide details in your response letter. – PeerJ Staff

Reviewer 1 ·

Basic reporting

Based on a set of polling communication station data sets, this paper uses one-dimensional convolutional neural network (CNN) to segment the data, thus improving its ability to identify various communication activities. This study evaluated the reliability of CNN in several real scenes, and checked its accuracy under noise interference, different intercepting signal lengths, interference at different frequency points, and dynamic changes in the position of outposts. The effectiveness and reliability of convolutional neural network in identifying communication behaviors in different environments are verified. The overall structure is relatively complete and the experimental design idea is rigorous, but there are some shortcomings as follows:
(1) In the background introduction of signal source, the introduction of each communication mode is slightly redundant. The main body of this paper is to verify the recognition ability of convolutional neural network for high similarity signals. However, the introduction of each communication mode in this paper is obviously irrelevant to the subject, so some appropriate simplification can be made in this part.
(2) Some experimental condition data settings in the article lack some proper explanations, for example, in the setting of noise conditions in the article, some numerical values are not explained, which may cause confusion to the readers, and we hope to add explanations.

Experimental design

In general, the experimental design is complete. However, considering the recognition rate of the convolutional neural network for the communication signal under different experimental conditions, and full experimental verification, but the problem of randomness of the experimental data is not well solved in the overall experimental design. The number of experiments should be appropriately increased to maintain the stability of the data. As seen from the data in Table 2, due to the randomness of the experimental results, leads to fluctuations in the experimental results. Although it does not affect the overall trend change, the method used in this article is to increase the number of experiments, and use the Monte Carlo idea to obtain stable values. But it is clear from the curve in Figure 6 that the number of experiments in this article is not enough to obtain stable values, so you can increase the number of experiments to obtain more accurate trend graphs.

Validity of the findings

no comment

Additional comments

no comment

Cite this review as

Reviewer 2 ·

Basic reporting

+ While the manuscript is free from grammatical errors, certain sections contain ambiguities that may confuse readers. For instance, in the last two sentences of the Introduction section, the first sentence discusses "The main goal", while the second sentence refers to "Our goal". This inconsistency could lead to confusion.
+ The relevant prior literature has been appropriately referenced throughout the manuscript.
+ Although the figures are relevant to the content of the manuscript and appropriately described and labeled, some figures suffer from poor quality, with text within them appearing unclear.
+ The topic presented in the manuscript matches the scope of the journal.

Experimental design

+ While the experimental design appears technically sound, there are some missing details. For instance, in the experiment about the impact of noise intensity, the authors stated the utilization of the Monte Carlo approach for conducting repeated recognition rate studies on various test sample sets. However, it remains unclear to me regarding the specific number of repetitions and the quantity of test samples involved.
+ Similarly, for the experiment about the impact of signal interception time, those details are also missing.

Validity of the findings

+ The observed trends in the results supports the validity the findings.

Additional comments

+ On page 8, "db" should be read as "dB".
+ This manuscript requires major revisions.

Cite this review as

---

## Round 0.2 · Minor Revisions

Please take all the comments from Reviewer 2 into account and revise the manuscript accordingly. In addition, please improve Figure 2, as its quality is too low.

Reviewer 1 ·

Basic reporting

The problems I concerned have been well addressed. I have no further questions.

Experimental design

no comment

Validity of the findings

no comment

Additional comments

no comment

Cite this review as

Reviewer 2 ·

Basic reporting

- The current version is better than the previous one.
- However, It is recommended that the Introduction section go into greater detail about the article's research focus and background, as well as explain why the research should be conducted. This aspect is still a little unclear.
- The Background section should provide a brief overview of the communication scenario in the article.

Experimental design

- A flowchart can be included in the Process of Communication Behavior Recognition section to help readers understand the entire signal recognition process.
- In the Data preprocessing section, it is recommended that the authors explain clearly how to segment the dataset based on images, as well as the benefits of doing so. That can help readers understand and be convinced.
- A more detailed description of the signal communication mode used in the article can be provided to help readers understand the signals and gain a better understanding of the experiments.
- The reasons for designing the convolutional kernel and pooling layer in CNN architecture should be explained.
- Please explain the time settings in Table 2.

Validity of the findings

No further comments

Additional comments

Although the manuscript has been improved, additional details are still required, as stated above.

Cite this review as

---

## Round 0.3 · accepted · Accept

We are pleased to announce that your manuscript has been accepted for publication following a thorough evaluation by two independent reviewers, both of whom have recommended acceptance.

Reviewer 2 ·

Basic reporting

No comment

Experimental design

No comment

Validity of the findings

No comment

Additional comments

The authors have addressed all of my concerns. I have no further comments.

Cite this review as